# Tumor-Specificity, Neurotoxicity, and Possible Involvement of the Nuclear Receptor Response Pathway of 4,6,8-Trimethyl Azulene Amide Derivatives

**DOI:** 10.3390/ijms23052601

**Published:** 2022-02-26

**Authors:** Kotone Naitoh, Yuta Orihara, Hiroshi Sakagami, Takumi Miura, Keitaro Satoh, Shigeru Amano, Kenjiro Bandow, Yosuke Iijima, Kota Kurosaki, Yoshihiro Uesawa, Masashi Hashimoto, Hidetsugu Wakabayashi

**Affiliations:** 1Faculty of Science, Josai University, Saitama 250-0295, Japan; sekaneko240240@gmail.com (K.N.); ty5yuta@gmail.com (Y.O.); takutakutakuwan1378@docomo.ne.jp (T.M.); hashi-m@josai.ac.jp (M.H.); hwaka@josai.ac.jp (H.W.); 2Research Institute of Odontology, Meikai University, Sakado, Saitama 350-0283, Japan; shigerua@dent.meikai.ac.jp; 3Division of Pharmacology, Department of Diagnostics and Therapeutics Sciences, Meikai University School of Dentistry, Saitama 350-0283, Japan; k-satoh@dent.meikai.ac.jp; 4Division of Biochemistry, Department of Oral Biology and Tissue Engineering, Meikai University School of Dentistry, Saitama 350-0283, Japan; kbando@dent.meikai.ac.jp; 5Department of Oral and Maxillofacial Surgery, Saitama Medical Center, Saitama Medical University, Saitama 350-0283, Japan; yoiijima@saitama-med.ac.jp; 6Department of Medical Molecular Informatics, Meiji Pharmaceutical University, Tokyo 204-8588, Japan; d196955@std.my-pharm.ac.jp

**Keywords:** 4,6,8-trimethyl azulene amide, oral cancer, tumor-specificity, neurotoxicity, QASR, molecular shape, caspase 3, cell cycle, NFκB, hormone receptor

## Abstract

Background: Very few papers covering the anticancer activity of azulenes have been reported, as compared with those of antibacterial and anti-inflammatory activity. This led us to investigate the antitumor potential of fifteen 4,6,8-trimethyl azulene amide derivatives against oral malignant cells. Methods: 4,6,8-Trimethyl azulene amide derivatives were newly synthesized. Anticancer activity was evaluated by tumor-specificity against four human oral squamous cell carcinoma (OSCC) cell lines over three normal oral cells. Neurotoxicity was evaluated by cytotoxicity against three neuronal cell lines over normal oral cells. Apoptosis induction was evaluated by Western blot and cell cycle analyses. Results: Among fifteen derivatives, compounds **7**, **9**, and **15** showed the highest anticancer activity, and relatively lower neurotoxicity than doxorubicin, 5-fluorouracil (5-FU), and melphalan. They induced the accumulation of a comparable amount of a subG_1_ population, but slightly lower extent of caspase activation, as compared with actinomycin D, used as an apoptosis inducer. The quantitative structure–activity relationship analysis suggests the significant correlation of tumor-specificity with a 3D shape of molecules, and possible involvement of inflammation and hormone receptor response pathways. Conclusions: Compounds **7** and **15** can be potential candidates of a lead compound for developing novel anticancer drugs.

## 1. Introduction

Azulene is a 10 π-electron non-benzenoid aromatic hydrocarbon with a fused structure of five- and seven-membered rings, showing a deep blue coloration. The resonance structure of azulene contains ionic cyclopentadienide and tropylium substructures, resulting in electrophilic substitution reactions at the 1- and 3-positions and nucleophilic addition reactions at the 2-, 4-, 6-, and 8-positions, along with the 2-position at the five-membered ring in some cases [1,2,3].

Azulene derivatives, including guaiazulene (Appendix A), are present in many plants and mushrooms, and applied as optoelectronic devices and ingredients used for hundreds of years in antiallergic, antibacterial, and anti-inflammatory therapies [4]. Several research studies have reported the applications of azulenes on oral diseases. For example, gargling with sodium azulene has been applied to maintain the compliance with afatinib treatment [5]. Azulene rinse has been applied to dry mouth and salivary gland dysfunction following radiotherapy, but with no convincing evidence [6]. Oral administration of marzulene (l-glutamine plus azulene) stimulates repair mechanisms of rat gastric mucosa after NaOH injury [7]. Guaizulene alleviated the paracetamol-induced glutathione (GSH) depletion and hepatic damage, possibly by its antioxidant activity [8]. Photodynamic activation of a lower concentration of guaiazulene suppressed inflammatory markers in peripheral blood mononuclear cells possibly by generating singlet oxygen [9]. On the other hand, very few reports of anticancer activity and quantitative structure–activity relationship (QSAR) analysis of azulenes have been reported [10,11,12].

In order to determine the tumor-specificity of azulene derivatives, we decided to use a set of four human oral squamous cell carcinoma cell lines and three human normal oral mesenchymal cells, rather than using human normal epithelial cells. This decision was based on our previous findings that (i) human normal epithelial cells (oral keratinocyte HOK, primary gingival epithelial cells HGEP) cannot be grown in regular culture medium (DMEM + 10%FBS) and, (ii) when HOK and HGEP were cultured in their specific growth factor-enriched medium, they began to rapidly grow like malignant cells, acquiring extremely higher sensitivity against anticancer drugs, resulting in the reduction of TS values below 1.0 [13].

We have previously investigated the anticancer activity of three groups of azulenes against human oral squamous cell carcinoma (OSCC) cell lines (Ca9-22, HSC-2, HSC-3, HSC-4) in comparison with three normal human oral mesenchymal cells (gingival fibroblast, HGF; periodontal ligament fibroblast, HPLF; pulp cell, HPC) (Appendix A): Ten alkylaminoguaiazulene derivatives (Appendix A) showed very weak tumor-specificity (TS = 1.1–2.3) [14]. Among ten guaiazulene amide derivatives (Appendix A), only one compound showed higher tumor-specificity (TS > 28.9) [15]. The tumor-specificity of all twenty-one azulene amide derivatives (Appendix A) were very weak (TS = 0.8~7.1) [16].

In continuation of finding more tumor-selective compounds, a total of fifteen 4,6,8-trimethyl azulene amide derivatives (Figure 1) were investigated for their cytotoxicity against four OSCC cell lines and three normal human oral mesenchymal cells. Since many cancer drugs have been reported to show severe neurotoxicity [17,18,19], neurotoxicity of fifteen 4,6,8-trimethyl azulene amide derivatives was also compared with those of popular anticancer drugs.

## 2. Results

### 2.1. Tumor-Specificity

Four human oral squamous cell carcinoma cell line (Ca9-22 originated from gingiva; HSC-2, HSC-3, HSC-4 from tongue) and three human mesenchymal normal oral cells (gingival fibroblast HGF, periodontal ligament fibroblast (HPLF) and pulp cells (HPC) were incubated for 48 h with various concentrations of compounds **1**–**15** and three reference compounds (doxorubicin (DOX), 5-fluorouracil (5-FU), melphalan (l-phenylalanine mustard, L-PAM) in DMEM supplemented with 10% fetal bovine serum (FBS) and antibiotics. Viable cell number was then determined by an MTT method in triplicate. Cytotoxicity of dimethyl sulfoxide (DMSO), used for dissolving these compounds, was subtracted. The cytotoxicity experiments were performed three times, and 50% cytotoxic concentration (CC_50_) was determined from the dose–response curve done in triplicate (Appendix A), and listed in Table 1. We have noticed that Ca9-22 (derived from gingiva) is more sensitive than other oral squamous cell lines (HSC-2, HSC-3, HSC-4) (derived from tongue) to most of the azulene compounds, and oral squamous cells are more sensitive than normal mesenchymal cells. Since such trends, however, are reproducible in three independent experiments (as shown in Appendix A), we presented the mean value.

When four cancer cells and three normal oral cells were used, compound **15** showed the highest tumor-specificity (TS = 17.9), followed by compound **7** (TS = 7.8) and compound **9** (TS = 5.7), higher than that of 5-FU (TS = 2.5) (D/B in Table 1). When two gingival tissue-derived cells Ca9-22 and HGF were used, compound **15** again showed the highest tumor-specificity (TS = 20.1), followed by compound **9** (TS = 10.2) and compound **7** (TS = 9.6) (C/A in Table 1).

For clinical application of candidate compounds, the database of their tumor-specificity and cytotoxicity against tumor cells provides useful information. Thus, PSE (Potency–Selectivity Expression (the ratio of the tumor-specificity to the CC_50_ against cancer cells) × 100) was calculated for all compounds. Compound **15** showed the highest value (PSE = 81.7; 101.0), followed by compound **9** (PSE = 17.3; 34.8) and compound **7** (PSE = 15.7; 23.1) (100D/B^2^; 100C/A^2^ in Table 1). PSE values of all derivatives were two-orders lower than that of doxorubicin (PSE = 12,325; 6270). However, PSE values of compounds **7**, **9**, and **15** were higher than that of 5-FU (PSE = 0.6; 2.7), and PSE value of compound **15** was higher than that of L-PAM (PSE = 56.5; 16.6). Based on these data, compounds **7**, **9**, and **15** were chosen for further analysis.

Dose–response curve of compounds **7**, **9**, and **15** and three reference compounds are shown in Figure 2. Compound **9** (B), doxorubicin (D) and melphalan (F) were cytotoxic, whereas compound **7** (A), compound **15** (C), and 5-FU (E) were cytostatic.

### 2.2. Apoptosis-Inducing Activity

Microscopical observation revealed that actinomycin D (1 μM) (B), a higher concentration of compound **7** (160 μM) (D), and low and high concentrations of compound **9** (60 and 120 μM) (E, F) induced cell shrinkage. On the other hand, compound **15** did not induce such morphological changes at 40 and 80 μM (G, H) (Figure 3).

Similarly, actinomycin D (1 μM), and higher concentration of compound **7** (160 μM), low and high concentrations of compound **9** (60 and 120 μM) increased the relative proportion of subG_1_ population (that reflect DNA fragmentation) to a nearly comparable level attained by actinomycin D. On the other hand, compound **15** at higher concentration (80 μM) only slightly increased subG_1_ population (*p* < 0.05) (Figure 4).

Western blot analysis demonstrated that compounds **7** and **9**, but not compound **15**, induced the cleavage of poly ADP-ribose polymerase (PARP) and procaspase 3, suggesting the induction of apoptosis (Figure 5). Since the extent of apoptosis induction was much lower than that induced by actinomycin D, the possibility of induction of other types of cell death such as necrosis has been suggested.

### 2.3. Neurotoxicity

Rat adrenal pheochromocytoma cell line (PC12), differentiated PC12 cells (Day 6) expressing characteristic neurites (dPC12), human neuroblastoma cell line (SH-SY5Y), and rat Schwann cell line (LY-PPB6) were treated for 48 h with various concentrations of test samples, and viable cell number was determined by MTT methods. Dose–response curve and CC_50_ values of three independent experiments are shown in Appendix A. The data were summarized in Table 2. Compounds **7** and **9** showed much lower neurotoxicity (NT = 1.4, 1.3, 1.0; 1.4, 1.0, 1.0) as compared with doxorubicin (NT = 70.6, 25.0, 44.9), 5-FU (NT = 11.0, 23.3, 3.4), and L-PAM (NT = 19.8, 7.2).

### 2.4. Computational Analysis

QSAR analysis of tumor-specificity of fifteen 4,6,8-trimethyl azulene amide derivatives (compounds **1**–**15**) were performed next. Significant correlation (*p* < 0.05) was found between the cytotoxicity against human oral squamous (OSCC) cell lines and 220 chemical descriptors (Appendix A). Six chemical descriptors that showed the highest correlation with the cytotoxicity against OSCC cells are shown in Figure 6. The cytotoxicity of 4,6,8-trimethyl azulene amide derivatives against human OSCC cell lines was correlated (r^2^ = 0.668~0.707, *p* = 0.0000853~0.00020) positively with AATSC0v (Topological Structure) (A), FASA_H (water accessible surface area) (C), GATS4s (Topological Structure) (E), and negatively with GATS1se (Topological Structure), FASA_P (water accessible surface area of all polar atoms) and vsurf_CW5 (Capacity factor) (Figure 6). These data suggest that the tumor-specificity of 4,6,8-trimethyl azulene amide derivatives are related to chemical descriptors that reflect 3D structure (AATSC0v, FASA_H, GATS4s, GATS1se, FASA_P vsurf_CW5) (Table 3).

### 2.5. Possible Nuclear Receptor/Stress Response Pathways

Nuclear receptors and stress response pathways that are possibly involved in the inhibition of OSCC growth by 4,6,8-trimethyl azulene amide derivatives were predicted. The specific cytotoxicity against OSCC cells were correlated with nuclear factor-kappa B (NFκB) agonist, estrogen receptor alpha with stimulator antagonist, thyroid stimulating hormone receptor (TSHR) antagonist, and glucocorticoid receptor agonist (Figure 7). These data suggest that the tumor-specificity of 4,6,8-trimethyl azulene amide derivatives might be coupled to the signaling pathway of NFκB and estrogen, thyroid stimulating hormone and glucocorticoid receptors.

## 3. Discussion

The present study demonstrated that tumor-specificity of fifteen 4,6,8-trimethyl azulene amide derivatives (mean TS value = 4.1) was generally lower than that of doxorubicin (TS = 35) (Table 1), confirming our previous papers (mean TS = 1.5) [13], 5.1 [14] and 2.0 [15]. Compounds **6**–**11** with alkyl groups showed a higher TS value (mean TS = 5.2) than compounds **1**–**5** with hydroxyalkyl groups (TS = 1.7) (Table 1). The size of side chain also may be the determinant of TS value: compounds **7**, **9**, and **15** having C4, C4, and C7 chain length showed the highest TS value (7.8, 5.7 and 17.9) (Table 1). This was supported by the QSAR analysis (Figure 6) that suggests the close association of TS value with chemical descriptors that reflect the 3D structure (AATSC0v, FASA_H, GATS4s, GATS1se, FASA_P vsurf_CW5) (Figure 6).

Dose–response curve demonstrated that compound **15** with the highest TS value inhibited the growth of Ca9-22 cells without complete killing of the cells (Figure 2C). Compound **15** induced trace amounts of a subG_1_ population (that reflects DNA fragmentation) (*p* = 0.029) (Figure 4) and caspase-3 activation (assessed by cleavage of PARP and procaspase 3) (Figure 5).

On the other hand, both compounds **7** and **9** were rather cytotoxic, inducing cell shrinkage (Figure 3C–F) and the accumulation of a subG_1_ population above 80 and 60 μM, respectively (*p* < 0.027) (Figure 2B), to a similar extent that was attained by actinomycin D (Figure 4). However, their caspase 3 activating activity was much lower than that of actinomycin D (Figure 5). This suggests the possibility that compounds **7** and **9** may induce different types of cell death from apoptosis.

Finally, a possible signaling pathway of 4,6,8-trimethyl azulene amide derivatives was estimated by Toxicity Predicators. Their tumor-specificity was matched at a higher probability with the onset of a signaling pathway of NFκB, and receptors for estrogen, thyroid stimulating hormone, and glucocorticoid (Figure 7). This is consistent with the reported anti-inflammatory activity of guaiazulene [4,9,20,21,22]. On the other hand, the involvement of other signaling pathways such as caspase and androgen receptor (Appendix A), transforming growth factor β (TGFβ), peroxisome proliferator activated receptor δ (PPARδ), endoplasmic reticulum stress (ER stress) response, and retinoid X receptor-α(RXR) (Appendix A) seems to be low. This further supports the non-apoptotic cell death induced by compounds **7, 9**, and **15**. It remains to identify the type of cell death induced by these compounds.

## 4. Materials and Methods

### 4.1. Materials

The following chemicals were obtained from the indicated companies: Dulbecco’s modified Eagle’s medium (DMEM) from Thermo Fisher Scientific (Waltham, MA, USA); FBS, 3-(4,5-dimethylthiazol-2-yl)-2,5-diphenyltetrazolium bromide (MTT), doxorubicin-HCl (DOX), melphalan (L-PAM) from Sigma-Aldrich (St. Louis, MO, USA); dimethyl sulfoxide (DMSO), actinomycin D from Fujifilm Wako Pure Chemical (Osaka, Japan); 5-FU from Kyowa (Tokyo, Japan); culture plastic dishes and 96-well plates from Techno Plastic Products (Trasadingen, Switzerland). Protease and phosphatase inhibitors were purchased from Roche Diagnostics (Basel, Switzerland).

### 4.2. Synthesis of Alkylamidoazulene Groups

*N*-(2-hydroxyethyl)-4,6,8-trimethylazulene-1-carboxamide (compound **1**), *N*-(3-hydroxypropyl)-4,6,8-trimethylazulene-1-carboxamide (compound **2**), *N*-(4-hydroxybutyl)-4,6,8-trimethylazulene-1-carboxamide (compound **3**), *N*-(5-hydroxypentyl)-4,6,8-trimethylazulene-1-carboxamide (compound **4**), *N*-(6-hydroxyhexyl)-4,6,8-trimethylazulene-1-carboxamide (compound **5**), *N*-propyl-4,6,8-trimethylazulene-1-carboxamide (compound **6**), *N*-butyl-4,6,8-trimethylazulene-1-carboxamide (compound **7**), *N*-pentyl-4,6,8-trimethylazulene-1-carboxamide (compound 8), *N*-hexyl-4,6,8-trimethylazulene-1-carboxamide (compound **9**), *N*-(2-methoxyethyl)-4,6,8-trimethylazulene-1-carboxamide (compound 10), *N*-(3-methoxypropyl)-4,6,8-trimethylazulene-1-carboxamide (compound **11**), *N*-(2-*N*′,*N*′-dimethylaminoethyl)-4,6,8-trimethylazulene-1-carboxamide (compound **12**), *N*-(3-*N*′,*N*′-dimethylaminopropyl)-4,6,8-trimethylazulene-1-carboxamide (compound **13**), *N*-(2-morpholinoethyl)-4,6,8-trimethylazulene-1-carboxamide (compound **14**) and *N*-(2-phenylethyl)-4,6,8-trimethylazulene-1-carboxamide (compound **15**) were synthesized by the reaction of trichloroacetyl derivatives and each amine at high yields, according to a previously published method [23,24,25].

### 4.3. Cell Culture

Human normal oral cells (HGF, HPLF, HPC) were established according to the guideline of intramural Ethic Committee (No. A0808) at 12–20 population doubling level (PDL) [26] and OSCC cell lines (Ca9-22, HSC-2, HSC-3, HSC-4), rat adrenal pheochromocytoma cell line (PC12), a human neuroblastoma cell line (SH-SY5Y), and rat Schwann cell line (LY-PPB6) (Riken Cell Bank, Tsukuba, Japan) were cultured at 37 °C in DMEM supplemented with 10% heat (56 °C, 30 min)-inactivated FBS, 100 U/mL, penicillin G and 100 µg/mL streptomycin sulfate under a humidified 5% CO_2_ atmosphere, as described previously [16].

Differentiated PC12 cells were prepared by the “overlay method”, as described previously. In short, PC12 cells were cultured in the serum-free DMEM supplemented with 50 ng/mL NGF, and at Day 3 overlayed with fresh NGF solution. The Day 6 cells with ex-tended neurites were used for the experiment.

### 4.4. Synthesis Assay for Cytotoxic Activity

Cells were inoculated at 2 × 10^3^ cells/0.1 mL in a 96-microwell plate. After 48 h, the medium was replaced with 0.1 mL of fresh medium containing different concentrations of test compounds. Control cells were treated with the same amounts of DMSO present in each diluent solution. Cells were incubated for 48 h and the relative viable cell number was then determined by the MTT method, as described previously [15]. We have prepared the controls that contain DMSO (1, 0.5, 0.25, 0.125, 0.063, 0.031, 0.016, 0.008%). Cytotoxicity induced by DMSO alone was subtracted from each well of a 96-microwell plate. The CC_50_ was determined from the dose–response curve of triplicate samples.

### 4.5. Calculation of Tumor-Selectivity Index (TS)

TS was calculated by dividing the mean CC_50_ (HGF + HPLF + HPC) by the mean CC_50_ (Ca9-22 + HSC-2 + HSC-3 + HSC-4), using seven cell lines ((D/B) in Table 1) [15], or dividing the CC_50_ (HGF) by the CC_50_ (Ca9-22) [(C/A) in Table 1], using two cell lines derived from the gingival tissue [27]. Normal keratinocytes, which are highly sensitive to many anticancer drugs [13,28], were not used in this study.

### 4.6. Calculation of Potency-Selectivity Expression (PSE)

PSE was calculated by dividing the TS value by the CC_50_ against tumor cells [15,29] [(D/B^2^) × 100 and (C/A^2^) × 100] (Table 1).

### 4.7. Western Blot Analysis

The cells were washed, lysed, and their protein extracts subjected to Western blot (WB) analysis, as described previously [30]. All protein samples of cell lysates (15 μg) were separated by SDS-PAGE using a Mini-Protean 3 Cell system (Bio-Rad Laboratories, Hercules, CA, USA). After electrophoresis, the separated proteins were transferred onto a PVDF filter using a Trans-Blot Turbo System (Bio-Rad Laboratories). The blots were blocked at room temperature for 50 min in skim milk (Morinaga-Nyugyo, Tokyo, Japan) and then probed for 120 min with a primary antibody cocktail (1:250) from Apoptosis Western Blot Cocktail kit (Abcam, Cambridge, UK). The blots were washed three times with Tris-buffered saline (pH 7.6) containing 0.05% Tween 20 and then probed for 90 min with a horseradish peroxidase-conjugated secondary antibody cocktail (1:100) from the kit. Immunoreactivities were detected using Amersham ECL Select (Cytiva, Tokyo, Japan). Images were acquired using ChemiDoc MP System (Bio-Rad Laboratories) and Image Lab 4.1 software (Bio-Rad Laboratories) [30].

### 4.8. Cell Cycle Analysis

Cells (approximately 10^6^ cells) were fixed with paraformaldehyde (Fujifilm Wako Pure Chemical) in PBS(−) and treated with ribonuclease (RNase) A (Sigma-Aldrich). After staining with propidium iodide (PI, Fujifilm Wako Pure Chemical) in the presence of 0.01% Nonidet-40 (Nacalai Tesque, Kyoto, Japan) to prevent cell aggregation, the cells were filtered through Falcon^®^ cell strainers (Corning Inc., Corning, NY, USA) and then subjected to cell sorting (SH800 Series, SONY, Tokyo, Japan), as described previously [16]. Cell cycle analysis was performed with Cell Sorter Software version 2.1.2. (SONY) [16].

### 4.9. Calculation of Chemical Descriptors

pCC50 (i.e., the −log CC_50_) was used for the comparison of the cytotoxicity between the compounds, since the CC_50_ values had a distribution pattern close to a logarithmic normal distribution. The mean pCC_50_ values for normal cells and tumor cell lines were defined as N and T, respectively [31]. Thus, T represents -log (mean CC_50_ against OSCC), N represents −log (mean CC_50_ against normal oral cells, T-N represents −log (mean CC_50_ for normal cells/mean CC_50_ for OSCC). The chemical structures were cleaned and standardized (removing salts and adjusting the protonation state (neutralize)). Then, the 3D-structure of each compound was generated by CORINA Classic (Molecular Networks GmbH, Nürnberg, Germany) and determined optimal 3D-structure with force field calculations (amber-10: EHT) in Molecular Operating Environment (MOE) version 2020.09 (Chemical Computing Group, Quebec, Canada). Chemical 2D and 3D descriptors were calculated with MOE version 2020.9 and Mordred version 1.2.0 (Python library) [32] based on optimal 3D structures. Descriptors that were duplicated and contained missing values and outliers were excluded from this analysis. Note that we regarded results outside 1st quartile −3 × interquartile (IQR) to 3rd quartile +3 × IQR range as outliers. Then, the multicollinearity of descriptors was analyzed. A threshold of 1 of the absolute value of the correlation coefficient was adopted as multicollinearity. When multicollinearity was detected, only descriptors that showed the highest correlation with objective variables among multicollinear descriptors was adopted. Finally, 1520 descriptors were used for this analysis.

### 4.10. Calculation of Chemical Descriptors

The activities against 59 signaling pathways [33], agonist and antagonist activities of the nuclear receptor, and stress response pathway were calculated by the chemical structures. In other words, all azulene derivatives were classified as positive or negative based on the calculated probabilities in Tox21 activity scores of 1 or higher for each signaling pathway using the Toxicity Predictor [33].

### 4.11. Statistical Analysis

Each experimental value was expressed as the mean ± standard deviation (SD) of triplicate or quadruplicate measurements. One-way ANOVA and Dunnett’s post-test were performed using IBM SPSS 27.0 statistics (IBM Co., Armonk, NY, USA). The correlation between chemical descriptors and cytotoxicity or tumor specificity was investigated using simple regression analyses by scikit-learn and SciPy with Python 3.8.5. Student’s *t*-test was performed using JMP Pro 6 (SAS Institute, Cary, NC, USA). The significance level was set at *p* < 0.05.

## 5. Conclusions

The present study demonstrated that three compounds from fifteen 4,6,8-trimethyl azulene amide derivatives selectively inhibited the growth of human oral squamous cell carcinoma cell lines. Their actions are either cytotoxic or cytostatic, accumulating subG_1_. The 3D-structure may be the determinant of tumor-selectivity. There was possible involvement of inflammation and hormone receptor pathway rather than caspase pathway in the selective cytotoxicity against OSCC cells. At present, TS values of compounds **7**, **9**, and **15** are not so high compared with anticancer drugs (Table 1). Since compound **9** was cytotoxic, further chemical modification of this compound that enhances the TS value should be done. On the other hand, compound **7**, compound **15**, and 5-FU showed cytostatic growth inhibition, and their action may be potentiated through combination with other types of anticancer drugs [34].

## Figures and Tables

**Figure 1 ijms-23-02601-f001:**
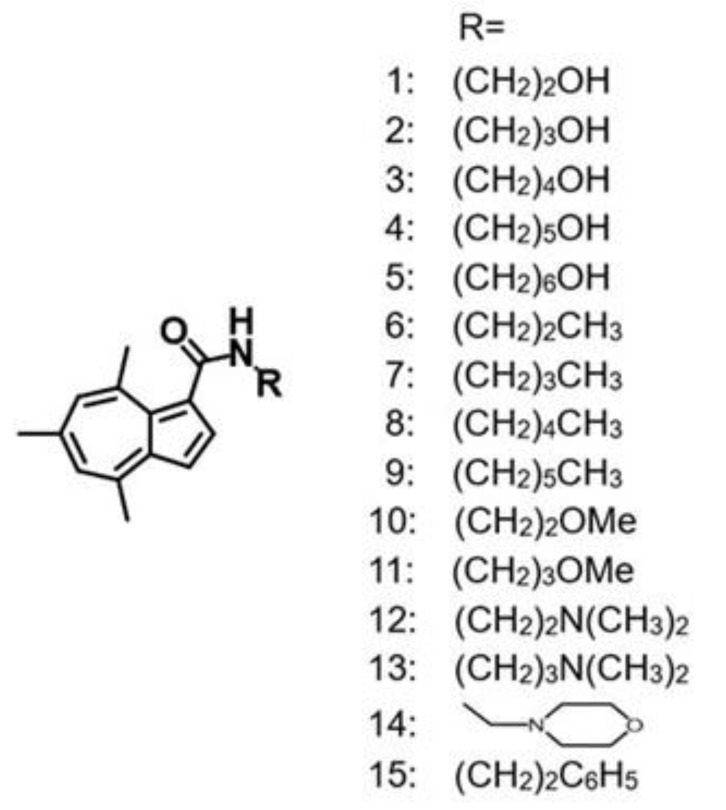
Structures of 4,6,8-trimethyl azulene amide derivatives.

**Figure 2 ijms-23-02601-f002:**
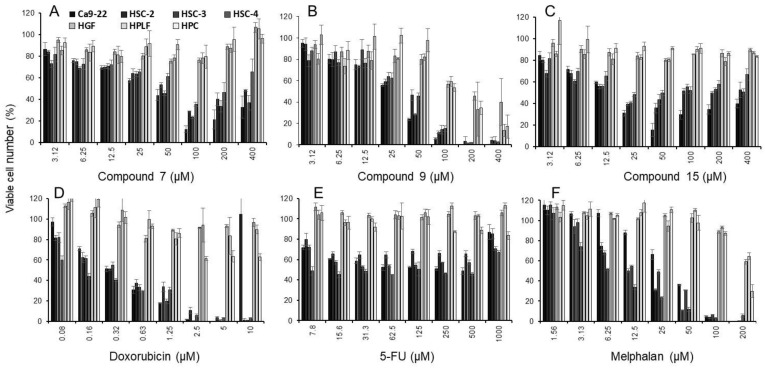
Dose–response curves of cytotoxicity induction by compound **7** (**A**), compound **9** (**B**), compound **15** (**C**), doxorubicin (**D**), 5-FU (**E**), and melphalan (**F**). Cells were incubated for 48 h with the indicated concentrations of compounds. Each value represents mean ±S.D. from three independent experiments which were done in triplicate. All data of the dose–response of fifteen 4,6,8-trimethyl azulene amide derivatives and three reference compounds (doxorubicin, 5-FU, melphalan) are available in Appendix A.

**Figure 3 ijms-23-02601-f003:**
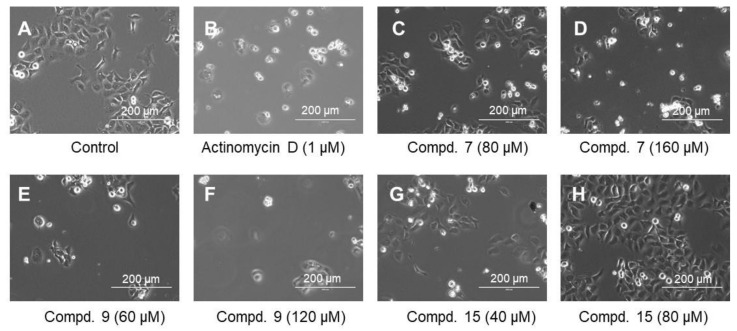
Morphological change induced in Ca9-22 cells after 24 h incubation without (control) (**A**), or with the indicated concentrations of actinomycin D (**B**), compound **7** (**C**,**D**), compound **9** (**E**,**F**) or compound **15** (**G**,**H**).

**Figure 4 ijms-23-02601-f004:**
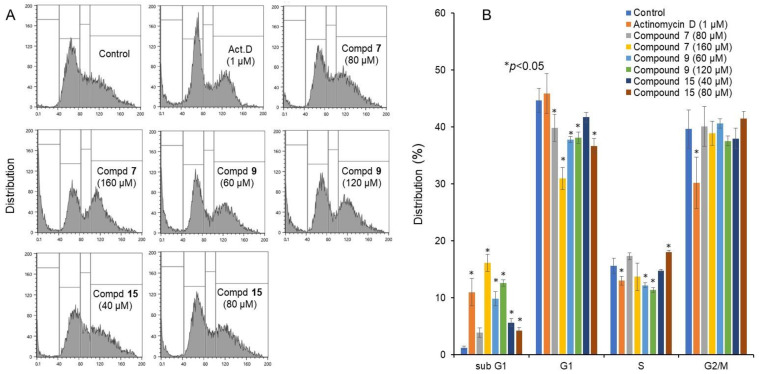
Cell cycle analysis of cytotoxicity induction by 4,6,8-trimethyl azulene amide derivatives. Ca9-22 cells were incubated for 24 h with the indicated concentrations of test samples and subjected to a cell sorter. (**A**) Representative cell cycle distribution pattern; (**B**) distribution into subG_1_, G_1_, S and G_2_/M phase. Each value represents the mean ±S.D. of triplicate determinations. * *p* < 0.05 between control and sample, examined by one-way ANOVA and Dunnett’s post-test.

**Figure 5 ijms-23-02601-f005:**
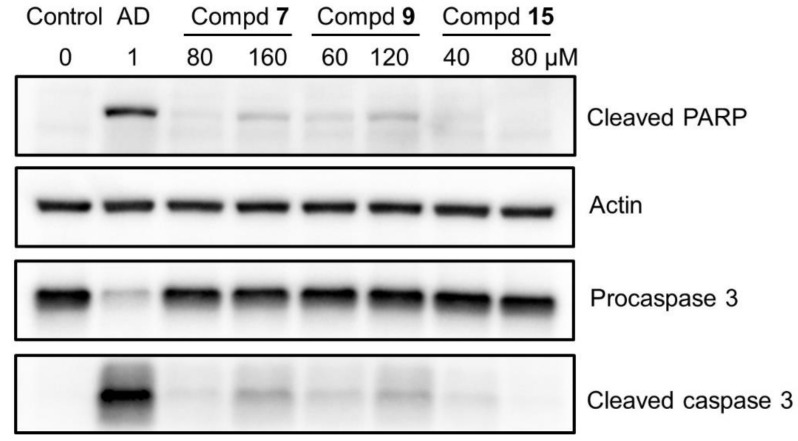
Western blot analysis of compounds **7**, **9**, and **15**. Ca9-22 cells were incubated for 24 h without (control) or with the indicated concentrations of test samples, and cell lysates were prepared as described in Materials and Methods. Protein (15 µg) was loaded to each lane for Western blot analysis, and the reacted bands were presented after contrast adjustment. Raw data of images after short, middle, and long exposures (without contrast adjustment) were shown in Appendix A.

**Figure 6 ijms-23-02601-f006:**
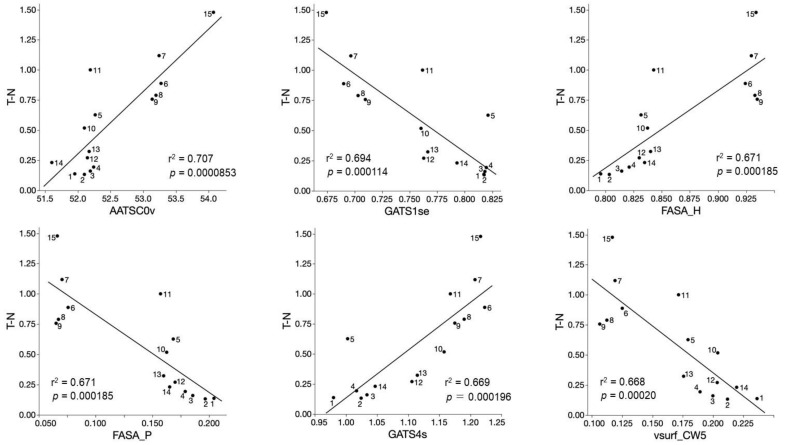
Top six chemical descriptors that showed higher correlation (r^2^ = 0.668~0.707) with tumor specificity of fifteen 4,6,8-trimethyl azulene amide derivatives. The mean negative log TS values (T-N) were plotted.

**Figure 7 ijms-23-02601-f007:**
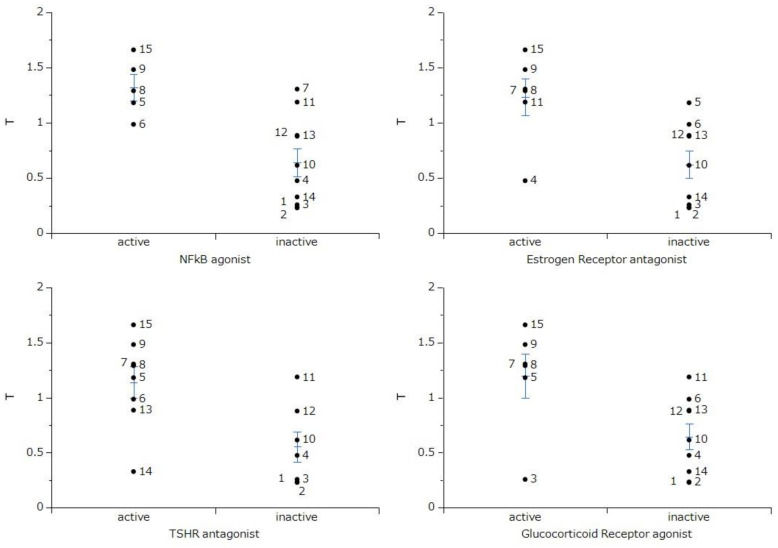
Nuclear receptor and stress response pathway involved in the selective toxicity against human oral squamous cell carcinoma cell lines. *t*-Test was performed on tumor cytotoxicity (T) between azulene derivatives predicted as active and inactive in each biochemical pathway. NFκB and TSHR indicate nuclear factor-kappa B and thyroid stimulating receptor, respectively.

**Table 1 ijms-23-02601-t001:** Cytotoxicity of 4,6,8-trimethyl azulene amide derivatives and reference compounds against human oral squamous cell carcinoma cell lines and human normal oral cells.

	CC_50_ (μM)												
	Human Oral Squamous Cell Carcinoma Cells					Human Normal Oral Cells							
	Ca9-22	HSC-2	HSC-3	HSC-4	Mean	HGF	HPLF	HPC	Mean	TS		PSE	
	(A)				(B)	(C)			(D)	(D/B)	(C/A)	100D/B^2^	100C/A^2^
1	335.1	400.0	397.4	400.0	383.1	400.0	400.0	400.0	400.0	1.0	1.2	0.3	0.4
2	379.3	377.8	400.0	400.0	389.3	400.0	400.0	400.0	400.0	1.0	1.1	0.3	0.3
3	233.4	400.0	400.0	377.8	352.8	400.0	400.0	400.0	400.0	1.1	1.7	0.3	0.7
4	288.1	313.8	361.0	375.6	334.6	400.0	391.6	376.9	389.5	1.2	1.4	0.3	0.5
5	53.5	40.8	63.2	106.5	66.0	293.3	363.9	181.9	279.7	4.2	5.5	6.4	10.2
6	71.1	86.9	59.0	196.9	103.5	400.0	400.0	400.0	400.0	3.9	5.6	3.7	7.9
**7**	**41.6**	**55.1**	**38.9**	**62.9**	**49.6**	**400.0**	**356.9**	**400.0**	**385.6**	**7.8**	**9.6**	**15.7**	**23.1**
8	46.7	52.4	46.3	60.1	51.4	354.9	266.8	328.0	316.6	6.2	7.6	12.0	16.2
**9**	**29.3**	**33.9**	**33.8**	**35.2**	**33.0**	**298.2**	**156.1**	**112.5**	**189.0**	**5.7**	**10.2**	**17.3**	**34.8**
10	124.2	337.0	181.3	328.7	242.8	400.0	400.0	400.0	400.0	1.7	3.2	0.7	2.6
11	55.9	76.5	46.3	81.6	65.1	400.0	355.6	400.0	385.2	5.9	7.2	9.1	12.8
12	110.7	162.7	101.3	155.6	132.6	303.2	270.7	168.0	247.3	1.9	2.7	1.4	2.5
13	120.5	173.1	73.6	153.6	130.2	327.7	293.0	201.6	274.1	2.1	2.7	1.6	2.3
14	391.1	377.8	309.4	400.0	369.6	400.0	400.0	400.0	400.0	1.1	1.0	0.3	0.3
**15**	**19.9**	**16.8**	**29.4**	**21.5**	**21.9**	**400.0**	**400.0**	**377.8**	**392.6**	**17.9**	**20.1**	**81.7**	**101.0**
DOX	0.4	0.2	0.4	0.1	0.3	10.0	9.0	10.0	9.7	34.5	25.1	12,325.0	6270.0
5-FU	191.2	703.0	673.5	33.1	400.2	1000.0	1000.0	1000.0	1000.0	2.5	5.2	0.6	2.7
L-PAM	34.7	11.9	20.1	6.9	18.4	200.0	200.0	173.4	191.1	10.4	5.8	56.5	16.6

Cells were incubated for 48 h with test compounds, and viable cell number was determined by MTT methods in triplicate. All dose–response curves of three independent experiments are shown in Appendix A. The CC_50_ (determined from the dose–response curves), TS, and PSE values are listed in Appendix A. DOX, doxorubicin; 5-FU, 5-fluorouracil; L-PAM, melphalan.

**Table 2 ijms-23-02601-t002:** Neurotoxicity of 4,6,8-trimethyl azulene amide derivatives and reference compounds.

	CC_50_ (μM)												
	Exp. 1				Exp. 2				Exp. 3				
	PC12	SH-SY5Y	LYPPB6	Mean	PC12	SH-SY5Y	LYPPB6	Mean	dPC12	Normal Cells	NT ^1^		
				(E)				(F)	(G)	(D)	(D/E)	(D/F)	(D/G)
**7**	59	400	400	286	68	400	400	289	400	386	1.4	1.3	1.0
**9**	35	73	56	55	36	63	74	58	19	189	3.5	3.3	10.2
**15**	378	42	400	273	400	400	400	400	400	393	1.4	1.0	1.0
DOX	0.07	0.02	0.32	0.14	0.26	0.12	0.78	0.39	0.22	10	70.6	25.0	44.9
5-FU	14	5.85	250	89.95	2	1	125	43	297	1000	11.0	23.3	3.4
L-PAM	N.D.	N.D.	N.D.	N.D.	2	2	25	10	26	191	N.D.	19.8	7.2

^1^ Neurotoxicity (NT) was determined by dividing the mean values of CC_50_ against normal human oral cells (D) by the mean of CC_50_ against PC12, SH-SY5Y and LYPPB6 (E in Exp. 1 and F in Exp. 2) or the CC_50_ against differentiated PC12 cells (G). Dose–response curve from which the CC_50_ values were delineated in Appendix A. N.D., not determined.

**Table 3 ijms-23-02601-t003:** Descriptors that significantly correlated with tumor-specificity (T–N) of fifteen 4,6,8-trimethyl azulene amide derivatives.

Important Descriptors	Significance of Correlation with T-N	Meanings
AATSC0v	r^2^ = 0.707; *p* = 0.0000853	Autocorrelation of Topological Structure of lag 0 atoms weighted by van der Waals volume
GATS1s	r^2^ = 0.694; *p* = 0.000114	Autocorrelation of Topological Structure of lag 1 atoms weighted by Sanderson EN
FASA_H	r^2^ = 0.671; *p* = 0.000185	Descriptor related to water accessible surface area of all hydrophobic atoms
FASA_P	r^2^ = 0.671; *p* = 0.000185	Descriptor related to water accessible surface area of all polar atoms
GATS4s	r^2^ = 0.669; *p* = 0.000196	Autocorrelation of Topological Structure of lag 4 atoms weighted by intrinsic state
vsurf_CW5	r^2^ = 0.668; *p* = 0.00020	Capacity factor

## Data Availability

Not applicable.

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
