# Peer review of "Tumor-Specificity, Neurotoxicity, and Possible Involvement of the Nuclear Receptor Response Pathway of 4,6,8-Trimethyl Azulene Amide Derivatives"

_ijms, 2022, doi:10.3390/ijms23052601_

Round 1

Reviewer 1 Report

In this article Naitoh et al have evaluated the anti-tumour potential of 15 azulene compounds with a focus of oral carcinoma. While the study is well performed, the authors can reconsider the controls used:

The authors have claimed that three azulene compounds are toxic specifically to tumours cell but not to normal ones. However, they have used tumour cell-lines of epithelial origin and normal cells of mesenchymal origin. It is necessary to re-evaluate the toxicity of these compounds using normal oral epithelial cells.

Author Response

Response to Reviewer 1’s comment

In this article Naitoh et al have evaluated the anti-tumour potential of 15 azulene compounds with a focus of oral carcinoma. While the study is well performed, the authors can reconsider the controls used:

Thank you for evaluating our paper.

The authors have claimed that three azulene compounds are toxic specifically to tumours cell but not to normal ones. However, they have used tumour cell-lines of epithelial origin and normal cells of mesenchymal origin. It is necessary to re-evaluate the toxicity of these compounds using normal oral epithelial cells.

Response

Thank you for nice suggestion. 

There was a big problem in using the epithelial normal cells for the initial screening of the most potent antitumor activity among fifteen 4,6,8-trimethyl azulene amide derivatives, due to the lack of appropriate positive control, as explained below.  In the present study, oral squamous cell carcinoma cell lines (Ca9-22, HSC-2, HSC-3, HSC-4) and mesenchymal normal oral cells (gingival fibroblast HGF, periodontal ligament fibroblast HPLF, pulp cell HPC), cultured in the same culture medium (DMEM + 10% FBS) were used for the treatment with samples to calculate their tumor-specificity (TS value).  This was based on our previous finding that (i) popular anticancer drugs, such as camptothecin, SN-38 (active principle of irinotecan), doxorubicin, daunorubicin, etoposide, mitomycin, 5-FU and docetaxel showed very high TS value, when mesenchymal normal oral cells were used (please see the second row from right, Table 1). However, if epithelial normal oral cells (HOK, HGEP) were used, in place of mesenchymal normal cells, under the same culture condition, their proliferation rapidly stopped and died out within 24 hours.  Thus, special culture media were required to keep their viability for longer culture:  keratin growth supplement (OKGS, Cat, No. 2652) for HOK cells (COSMO BIO Co/Ltd.); CnT-PR medium  (CELLnTEC Advanced Cell Systems AG) for HGEP cells.  Both HOK and HGEP cultured in such growth factor-enriched special media were found to be much more sensitive to doxorubicin, daunorubicin, etoposide, mitomycin, 5-FU, melphalan, gefinitib than human oral squamous cell carcinoma cell lines, reducing the TS value below 1 in most cases (rightmost column in Table 1). It is apparent that CC50 of doxorubicin against oral squamous cell carcinoma and epithelial normal cells were one or two orders higher than that of normal mesenchymal cells, regardless of cell density (Figure 2F). 

Table 1 Cytotoxic activity of anticancer drugs against human oral malignant

Each value represents the mean of triplicate determinations. HGF, Human gingival fibroblast; HPC, human pulp cells; HPLF, human periodontal ligament fibroblast; Ca9-22 (derived from gingival tissue), HSC-2, HSC-3 and HSC-4 (derived from tongue), human oral squamous cell carcinoma cell lines; CC50, 50% cytotoxic concentration; CPT, camptothecin; DXR, doxorubicin; DNR, daunorubicin; ETP, etoposide; MMC, mitomycin C, 5-FU, 5-fluorouracil; DOC, docetaxel. cited from; Sakagami et al., Anticancer Res 17, 1023-1030, 2017

Figure 2.  Keratinocyte toxicity induced by doxorubicin did not depend on the cell density. Adherent Ca9-22 (A), HSC-2 (B), HOK (C), HGF (D) and HPLF (E) cells were inoculated at very low (LL), low (L), high (H) or very high (HH) cell density, incubated for 48 h with the indicated concentrations of doxorubicin, and then the relative viable cell number (absorbance at 540 m) was determined by MTT method to calculate the 50% cytotoxic concentration (CC50). Each value represents mean ±S.D. of triplicate determinations. Insert (F): CC50 was plotted as a function of cell density (absorbance at 540 nm) of control cells at the time of cell harvest.   cited from; Sakagami et al., Anticancer Res 17, 1023-1030, 2017

Furthermore, doxorubicin induced apoptotic cell death of HOK in the special medium, characterized by disappearance of cell surface microvilli (A) and caspase activation (assessed by cleavage of PARP and procaspase 3) (B) in Figure 3.  Possibly, growth factor(s) present in the special medium for keratinocyte might have stimulated the proliferation of HOK and HGEP, thus conferring them the skyrocketed sensitivity to anticancer drugs. Further experiments are needed to elucidate this mechanism.  Based on these our original data, a set of epithelial cancer cells and mesenchymal normal cells, rather than a set of both epithelial cells, were used in this paper.

Figure 3. Doxorubicin induced apoptosis in HOK human normal oral keratinocyte. (A) Transmission electron microscope (TEM) analysis.   HOK cells were treated for 0 (a) or 24 (b, c) h with 10 μM doxorubicin and fine cell structure of adherent (a, b) and detached cells (c) were observed under TEM.  Bars indicate 1 μm (a, b) and 2 μm (c). (B) Western blot analysis. HOK cells were treated for 24 h with either vehicle (DMSO) (0.2%) (lane 1), doxorubicin (0.4 and 2 μM) (lane 2 and 3) or actinomycin D (1 μM) (lane 4). Detached and attached cells were combined for western blot analysis. 

Since there is no appropriate positive control, we did not investigate whether compounds 7,9, and 15 shows any antitumor potential using oral keratinocyte. However, when optimal condition for culturing oral keratinocyte were once established, we will perform additional experiment you have suggested.   Thank you for your precious comment that gives us new direction of research. 

Thus, we have added the following sentences in the 3rd paragraph of introduction (page 3):

“In order to determine the tumor-specificity of azulene derivatives, we decided to use a set of four human oral squamous cell carcinoma cell lines and three human normal oral mesenchymal cells, rather than using human normal epithelial cells.  This decision was based on our previous findings that (i) human normal epithelial cells (oral keratinocyte HOK, primary gingival epithelial cells HGEP) cannot be grown in regular culture medium (DMEM+10%FBS) and (ii) when HOK and HGEP were cultured in their specific growth factor-enriched medium, they began to rapidly grow like malignant cells, and acquiring extremely higher sensitivity against anticancer drugs, resulting in the reduction of TS values below 1.0 [13]”

Other corrections

Since the location of the break in the boundary region of the cell cycle was wrong, the proportion of the distribution to each phase of cell cycle was recalculated.  We replaced old Figure 4 A and B by revised new ones.  Accordingly, explanations of this figure in the text were also corrected as described below.  Please be noted that statements of sbG1 population were not changed.  Sorry for our mistake. 

Page 5. Line155-157. " On the other hand, compound 15 at higher concentration (80 μM) only slightly increased subG1 population (p<0.05) and G2/M phase cells (p=0.092) (Figure 4)."  →   "and G2/M phase cells (p=0.092)" deleted. (line 165)

Page 9. Line 246-247.  "Compound 15 induced trace amounts of subG1 population (that reflects DNA fragmentation) (p=0.029) and G2/M arrest (p=0.092) (Figure 4) and
caspase-3 activation (assessed by cleavage of PARP and procaspase 3) (Figure 5)."  → "and G2/M arrest (p=0.092)" deleted. (line 254-256)

Page 12. Line 395. "Their actions are either cytotoxic or cytostatic, accumulating subG1 or G2/M.” →  " or G2/M"  deleted  (line 403)

The order of references was changed, and one new reference (Ref. 34) was added, as indicated by red.  

Reviewer 2 Report

In this manuscript, the authors synthesized fifteen 4,6,8-Trimethyl azulene amide derivatives and their anticancer activity was evaluated. Herein, they tested the tumor-specificity against four cancer cell lines over three normal oral cells. Besides, they also characterized the neurotoxicity and apoptosis induction.

  1. In table 1, they calculated the tumor specificity using the mean of CC50 of several different cell lines. While the standard deviation of these cell lines are quite big. Is this a good idea to only get the mean value and conclude the result?
  2. They mentioned that cytotoxicity of dimethyl sulfoxide (DMSO), used for dissolving these compounds, was subtracted. How many percentages of DMSO was used for dissolving these compounds? If it is high percentage, is there a potential problem for treatment purposes?
  3. On line 120, there is a typo error.
  4. Based on the results shown in Figure 2, they found out that Compound 9, doxorubicin and melphalan were cytotoxic. Compound 7, compound 15 and 5-FU were cytostatic. How can this information help us regarding the future application?
  5. There are many grammar and typo issues throughout the whole manuscript.

Author Response

Response to Reviewer 2’s comment

In this manuscript, the authors synthesized fifteen 4,6,8-Trimethyl azulene amide derivatives and their anticancer activity was evaluated. Herein, they tested the tumor-specificity against four cancer cell lines over three normal oral cells. Besides, they also characterized the neurotoxicity and apoptosis induction.

  1. In table 1, they calculated the tumor specificity using the mean of CC50 of several different cell lines. While the standard deviation of these cell lines are quite big. Is this a good idea to only get the mean value and conclude the result?

Response

Yes, I think we can use the mean value. 

We have noticed that Ca9-22 (derived from gingiva) is more sensitive than other oral squamous cell lines (HSC-2, HSC-3, HSC-4) (derived from tongue) to most of azulene compounds, and oral squamous cells are more sensitive than normal mesenchymal cells. Since such trends, however, are reproducible in three independent experiments (as shown in supplementary data Figure S2, S3 and S4), we presented the mean value.  This statement was included in the result section (line 102-107).

  1. They mentioned that cytotoxicity of dimethyl sulfoxide (DMSO), used for dissolving these compounds, was subtracted. How many percentages of DMSO was used for dissolving these compounds? If it is high percentage, is there a potential problem for treatment purposes?

Response

We have prepared the controls that contain DMSO (1, 0.5, 0.25, 0.125, 0.063, 0.031, 0.016, 0.008%).  Cytotoxicity induced by DMSO alone was subtracted from each well of 96-microwell plate.  This statement was incorporated into the text (line 322-324).

  1. On line 120, there is a typo error.

Response

Thank you for your careful check.

“Compound 15 (B)” was corrected to “Compound (C)”. (line 143)

  1. Based on the results shown in Figure 2, they found out that Compound 9, doxorubicin and melphalan were cytotoxic. Compound 7, compound 15 and 5-FU were cytostatic. How can this information help us regarding the future application?

Response

Thank you for nice comment.

At present, TS values of compounds 7, 9 and 15 are not so high as compared with anticancer drugs (Table 1).  Since compound 9 was cytotoxic, further chemical modification of this compound that enhance the TS value will be done.  On the other hand, compound 7, compound 15, and 5-FU showed cytostatic growth inhibition, and their action will be potentiated by combination with other types of anticancer drugs [34].  This statement was added to the conclusion section (line 406-411.

  1. There are many grammar and typo issues throughout the whole manuscript.

Response

We have checked typographical errors. 

Other corrections

Since the location of the break in the boundary region of the cell cycle was wrong, the proportion of the distribution to each phase of cell cycle was recalculated.  We replaced old Figure 4 A and B by revised new ones.  Accordingly, explanations of this figure in the text were also corrected as described below.  Please be noted that statements of sbG1 population were not changed.  Sorry for our mistake. 

Page 5. Line155-157. " On the other hand, compound 15 at higher concentration (80 μM) only slightly increased subG1 population (p<0.05) and G2/M phase cells (p=0.092) (Figure 4)."  →   "and G2/M phase cells (p=0.092)" deleted. (line 165)

Page 9. Line 246-247.  "Compound 15 induced trace amounts of subG1 population (that reflects DNA fragmentation) (p=0.029) and G2/M arrest (p=0.092) (Figure 4) and
caspase-3 activation (assessed by cleavage of PARP and procaspase 3) (Figure 5)."  → "and G2/M arrest (p=0.092)" deleted. (line 254-256)

Page 12. Line 395. "Their actions are either cytotoxic or cytostatic, accumulating subG1 or G2/M.” →  " or G2/M"  deleted  (line 403)

The order of references was changed, and one new reference (Ref. 34) was added, as indicated by red.  

Round 2

Reviewer 1 Report

The authors have addressed my concerns.